# PrEP in the key population community: A qualitative study of perspectives on pre-exposure prophylaxis by gay, bisexual and other men who have sex with men and female sex workers in Kigali, Rwanda

Jonathan Ross[1]*, Josephine Gasana[2], Natalia Zotova[1], Giovanni Ndabakuranye[2], Fabiola Mabano[2], Charles Ingabire[2], Adebola Adedimeji[3,4], Gad Murenzi[2], Viraj V. Patel[1]

1 Department of Medicine, Albert Einstein College of Medicine, Bronx, New York, United States of America, 2 Einstein-Rwanda Research and Capacity Building Program, Research for Development (RD Rwanda) and Rwanda Military and Referral Teaching Hospital, Kigali, Rwanda, 3 Department of Epidemiology and Population Health, Albert Einstein College of Medicine, Bronx, New York, United States of America, 4 Departments of Social Sciences and Health Policy and Implementation Science, Wake Forest University, Winston-Salem, North Carolina, United States of America

* joross@montefiore.org

## Abstract

Gay, bisexual and other men who have sex with men (GBMSM) and cisgender female sex workers (FSW) are increasingly and disproportionately impacted by HIV in sub-Saharan Africa, yet current PrEP care models in this region are not optimized for these communities. Limited data exist describing experiences and preferences of GBMSM and FSW with respect to accessing and using PrEP. We conducted qualitative, semi-structured interviews with GBMSM and FSW recruited from three health centers and seven community organizations in Kigali, Rwanda. Data were analyzed using a mixed deductive and inductive approach to describe key themes related to initiating and adhering to PrEP. Participants included 18 GBMSM and 14 FSW; 12 were using PrEP at the time of interview, 9 had previously used PrEP, and 11 had never used it. Participants highlighted the central role of their social networks as key sources of information about and support for PrEP use, and described a strong motivation to use PrEP as a way to protect both themselves and their communities from HIV. While stigma and discrimination were pervasive, these were experienced differently by GBMSM and FSW. Participants suggested community access points that allowed more discreet and less frequent contact with health care workers as important and desired strategies to improve engagement. These findings suggest that leveraging community resources for disseminating information about HIV prevention and delivering PrEP could contribute to successful implementation of PrEP for GBMSM and FSW in Rwanda and other settings in SSA.

**Data availability statement:** Full transcripts are not publicly available as they contain sensitive information, which, though de-identified, include personal narratives that could result in identification of individuals, and because permission for public dissemination of raw data was not solicited from participants at the time of informed consent, nor from the Rwanda National Ethics Committee. De-identified data that support the findings of this study may be made available for researchers who meet the criteria for access to confidential data upon reasonable request. For data inquiries, please contact Ms. Valentine Ingabire, secretary of the Rwanda National Ethics Committee, at info@rnecrwanda.org.

**Funding:** The authors did not receive specific funding for this study. However, the work was supported through program grants from the U.S. National Institutes of Health's National Institute of Allergy and Infectious Diseases, the Eunice Kennedy Shriver National Institute of Child Health and Human Development, the National Cancer Institute, the National Institute of Mental Health, and the National Institute on Drug Abuse, as part of Central Africa IeDEA (U01 AI096299); and by the Einstein-Rockefeller-CUNY Center for AIDS Research (P30-AI124414), which is supported by the following NIH Co-Funding and Participating Institutes and Centers: NIAID, NCI, NICHD, NHBL, NIDA, NIMH, NIA, FIC, and OAR. The content is solely the responsibility of the authors and does not necessarily represent the official views of the National Institutes of Health. The funders had no role in study design, data collection and analysis, decision to publish, or preparation of the manuscript.

**Competing interests:** The authors have declared that no competing interests exist.

## Background

Nearly three-quarters of the 34 million people living with HIV reside in sub-Saharan Africa (SSA) [1]. Gay, bisexual and other men who have sex with men (GBMSM) and cisgender female sex workers (FSWs) are among the key populations (KPs) who are at the highest risk of HIV, and are increasingly and disproportionately represented among new HIV infections in SSA and globally [2–4].

Pre-exposure prophylaxis (PrEP) is highly effective in preventing HIV [5–8]. In its 2016 guidelines, the World Health Organization recommended PrEP for all individuals at substantial risk of HIV [9]. Currently, more than half of global PrEP users reside in SSA, where over 1 million individuals received a prescription in 2022 [10]. While most national PrEP programs in SSA prioritize KPs [11], current care models are not optimized for them. For example, in Rwanda, PrEP is available in primary health centers, settings where key populations may face discrimination and stigma [12]. Thus, despite high willingness to use PrEP among GBMSM and FSW [13–17], engagement remains low [18–20]. Evidence to date suggests that individual (e.g., health literacy, physical and sexual violence, stigma), institutional (e.g., discrimination in health settings), and structural factors (e.g., poverty, anti-homosexuality legislation) impact the opportunity cost around accessing health services and contribute to limited PrEP use [21–25].

Rwanda is a landlocked country situated in central Africa with a total population of 13 million; the largest and capital city is Kigali, with an approximate population of 1.8 million [26]. A majority of the estimated 14,000 FSW (2021 population size estimate [27]) and 18,000 GBMSM (2023 population size estimate [28]) in Rwanda live in Kigali, where the HIV prevalence is higher (4.3%) than the national prevalence of 3% [29]. HIV prevalence among KPs in Rwanda is markedly higher, estimated at 46% among FSW in Kigali and 7% among GBMSM [30]. Rwanda decriminalized same-sex relationships as well as sex work in 2018, but does not have legal recognition for transgender persons. Nonetheless, there are no anti-discrimination laws specifically protecting lesbian, gay, bisexual transgender and intersex (LGBTI) persons, and same-sex marriages are not recognized. Overall, Rwandan society remains socially conservative, with relatively low support for LGBTQ rights [31].

In February 2019, the government of Rwanda began implementing oral PrEP for KPs as part a comprehensive HIV prevention package [32]. Initial efforts in Rwanda focused on FSW, GBMSM and serodifferent couples; more recently efforts have been made to expand PrEP to adolescent girls and young women, index sex partners and individuals in the general population with considerable risk of HIV acquisition. Nonetheless, few data exist on experiences and practices of Rwandan FSW and GBMSM with respect to accessing and using PrEP. To explore these relationships, we conducted a qualitative study to understand barriers to, facilitators of and preferences for PrEP use among Rwandan GBMSM and FSW.

## Methods

### Ethics statement

The Rwanda National Ethics Committee (RNEC approval number 700/RNEC/2021) and the Institutional Review Board of the Albert Einstein College of Medicine (2020–12619) approved the study, which was conducted according to the principles expressed in the Declaration of Helsinki and is reported in accordance with Consolidated Criteria for Reporting Qualitative Research (COREQ) guidelines [33]. Written informed consent forms were also obtained from all participants prior to study enrollment. The informed consent forms were kept in a locked office, transcripts were de-identified, and audio-recordings were destroyed after transcription.

### Study setting and population

PrEP implementation in Rwanda has been rolled out in stages, with a focus on key populations as groups at higher risk of acquiring HIV, particularly FSW and GBMSM. By the end of June 2023, the number of FSW and GBMSM initiating PrEP medication had gradually increased from 10,078 in July 2022–10,789 in June 2023 [34]. While national data on retention are currently unavailable, early studies have demonstrated high PrEP persistence among both GBMSM and FSW [35,36]. Entry points into PrEP include self-referral, HIV testing sites, family planning services, STI treatment settings and referral from community-based organizations. PrEP care occurs at primary health centers, primarily within HIV clinics and is not differentiated for key populations

For the current study, we recruited participants from three health centers in Kigali (Busanza, Remera and Gikondo), as well as from seven community KP associations of GBMSM and FSW that operate in Kigali and with whom our research group has an ongoing academic-community collaboration. The participating health centers are considered "friendly" and less stigmatizing by many GBMSM in Kigali and have fairly robust PrEP programs.

### Participant recruitment

To recruit participants for the study, research staff visited health centers and KP associations to present the study to providers, solicit input on data collection instruments, and request assistance in identifying individuals who were on PrEP, who had stopped PrEP, and who had been evaluated for but never initiated PrEP. Providers at the three health centers and at the KP associations were asked to provide brief information about the study to potential participants, and refer individuals who expressed interest in participation to study staff for eligibility screening. We included participants who: 1) were HIV-negative by self-report; 2) self-identified as GBMSM or FSW; 3) were at least 18 years of age. Individuals who were unable to communicate in Kinyarwanda or unable to provide informed consent were excluded. We purposefully recruited individuals who were using PrEP at the time of the interview, had previously used PrEP but were not using it at the time of the interview, and individuals who had never used PrEP, such that the sample was approximately evenly divided between these categories. Participants received a stipend of 10,000 RWF (approximately $10) for their time. Recruitment began on June 8, 2022 and ended on July 22, 2022.

### Data collection

We developed a semi-structured interview guide informed by the socio-ecological model (S1 Text). The interview guide explored individual and structural barriers to and facilitators of PrEP engagement, as well as preferences for PrEP care delivery, using open-ended questions as well as specific probes to explore areas of particular interest. All participants provided written informed consent prior to enrolling in the study and being interviewed. Interviews lasting approximately 60–90 minutes were conducted in Kinyarwanda language (the language most commonly spoken in Rwanda) by four Rwandan research staff including one psychologist (CI) and three social workers (JG, GN, FM), all with multiple years' clinical and qualitative research experience working with GBMSM and FSW living with and at risk for HIV, and who

received specific training on the interview guide developed for this study. Interviewers had no prior relationship with participants.

All interviews were conducted in a private room with the presence of study staff only, to ensure recording quality and participant privacy. Each interview was conducted by a team of two interviewers, where one led the interview and the other took field notes. Interview quality was monitored by CI, observing early interviews and providing feedback with the data collection team and JR (principal investigator) through weekly conference calls. Transcripts were reviewed and compared to field notes to ensure that they reflected all content that arose during interviews. Interview guides were iteratively refined to clarify and further explore emerging themes relevant to implementation of PrEP uptake and retention in Rwanda.

### Data analysis

Audio recordings were transcribed and translated into English transcripts, which were then analyzed using both inductive and deductive thematic analysis approach to describe key barriers, facilitators and preferences. Our approach was deductive in that the investigative team developed the initial coding scheme using the socioecological model to categorize common themes, and was inductive in that we iteratively refined the codebook based on emergent themes from reading the first six transcripts. Discrepancies were discussed and resolved by consensus. Using DeDoose software (20), the final coding scheme was independently applied to all interviews by 4 coders (CI, JG, GN, FM), with each interview being coded by at least 2 investigators. The coding team regularly reviewed progress and discussed issues that arose, resolving them by consensus. These discussions also helped to establish that saturation had been reached. After all interviews were coded, excerpts were reviewed, examining themes within each code as well as between codes and using the constant comparative method to identify, refine and consolidate emergent themes. Specifically, the analytic team examined "repeating ideas" within each code to identify emergent themes. Emergent themes were entered into a matrix, looking specifically at barriers to, facilitators of and preferences for PrEP use at different levels of the SEM. Throughout the analysis phase, the team regularly met to discuss and achieve consensus on emerging themes.

### Results

From June–July 2022, we interviewed 32 participants. The median age of participants was 31; 18 (56%) were GBMSM and 14 (44%) were FSW. Overall, 12 participants were using PrEP at the time of interview, 9 had previously used PrEP, and 11 had never used PrEP (Table 1). Analysis revealed four major themes: (1) social networks as key sources of PrEP information and support; (2) PrEP as protective to individuals and to the community; (3) multiple, but differing, stigmas as barriers to PrEP engagement; and (4) community access as an important strategy to improve PrEP engagement.

### Social networks as key sources of PrEP information and motivation

Overall, participants were familiar with PrEP, although many reported that overall knowledge about it in Rwandan society, and even among the communities of GBMSM and sex workers, was lacking.

For both GBMSM and FSW, social networks, including friends, peers, community mobilizers, and KP associations, emerged as key sources of PrEP information, mediated through the trust, acceptance and lack of stigma in these relationships. At the interpersonal level, participants described a high degree of trust in information that came from friends and peers; individuals who used PrEP felt comfortable sharing information within their communities about its benefits as well as how to access it. The potential for friends and peers to also benefit from PrEP was described as a strong motivation to share information widely, highlighting the key role of these networks as a source of counseling and information. However, many were hesitant to share information about PrEP more broadly because of the potential discrimination they might experience.

*"I knew it from others because sometimes we talk as friends. So, my friends told me some who are [living with HIV] and those who are [HIV negative]; I used to think that only [people living with HIV] have ART to take but my friends told*

**Table 1. Participant characteristics.**

| | KP Category | |
|---|---|---|
| | **FSW (N = 14)** | **GBMSM (N = 18)** |
| **Age categories** | | |
| <25 years | 4 (29%) | 5 (28%) |
| 25–30 years | 2 (14%) | 7 (39%) |
| >30 years | 8 (57%) | 6 (33%) |
| **Education Level** | | |
| No school or primary school only | 9 (64%) | 2 (11%) |
| Some secondary school | 4 (29%) | 11 (61%) |
| University | 1 (7%) | 5 (28%) |
| **PrEP use** | | |
| Current | 6 (42%) | 6 (33%) |
| Prior | 3 (21%) | 6 (33%) |
| Never | 5 (37%) | 6 (34%) |
| **Sex at birth** | | |
| Female | 14 (100%) | 0 (0%) |
| Male | 0 (0%) | 18 (100%) |
| **Sexual Partners** | | |
| Men | 14 (100%) | 12 (67%) |
| Men and Women | 0 (0%) | 6 (33%) |
| **Number of sexual partners in past 6 months** | | |
| 1–10 | 1 (7%) | 17 (94%) |
| 11–20 | 4 (29%) | 1 (6%) |
| >20 | 9 (64%) | 0 (0%) |
| **STI diagnosis in prior 6 months** | 7 (50%) | 2 (11%) |
| **Exchange sex for money in prior 6 months** | 14 (100%) | 6 (33%) |
| **Reported challenges related to condom use during sex** | | |
| No challenges | 5 (36%) | 9 (50%) |
| Sexual partners refuse condoms because we use PrEP | 0 (0%) | 1 (6%) |
| Condoms feel uncomfortable | 8 (57%) | 3 (17%) |
| Condoms reduce pleasure | 1 (7%) | 5 (28%) |
| **Time since most recent HIV test** | | |
| ≤1 month | 1 (7%) | 7 (39%) |
| 2–4 months | 8 (57%) | 8 (44%) |
| ≥5 months | 4 (29%) | 2 (11%) |

me that also [people who are HIV-negative] take PrEP due to the fact that some engage in commercial sex which can result into HIV infection. For that reason, I approached a health center for details and the medicine as well, so that I can take it."

*- 31-year-old female, currently on PrEP*

"I shared it with many people. Most of the time it was to encourage them by saying, 'It would be great if you take it.'…I disclosed it to our community members because it felt normal. I did not share it with non-members of our community.

*- 32-year-old male, previously on PrEP*

Although most participants felt that the information acquired through social networks was accurate, some described how incorrect (e.g., regarding cost) or negative (e.g., emphasizing side effects) information circulating in these networks discouraged some individuals from engaging in PrEP. One participant described misinformation about uncommon medication side effects:

*"There exist a lot of rumors that the medicine causes acne and so on. A friend of mine told me that it has caused him facial acne… though I haven't seen it yet - just rumors - but it's said that it causes obesity and heart-related effects."*

*- 33-year-old male, currently on PrEP*

At the institutional level, health care providers, including physicians, nurses and community health workers, were felt by most participants to be trustworthy sources of information about PrEP. These sources were particularly important for participants who described difficulty accessing accurate PrEP information through more informal means or who already had an established connection to health centers. As one sex worker explained:

*We went to there for general treatment, then we heard doctors who were teaching and we were curious to listen to what they were teaching about; we came to find that it was about this PrEP. Then we said, 'Maybe this is helpful. Let's proceed.' We approached the nurse, she fully explained it to us, she filed us, and she started giving us the medicine. She helped us.*

*- 25-year-old female, currently on PrEP*

However, for some participants, particularly GBMSM, stigmatizing attitudes from health care providers overshadowed the perceived and actual benefits of learning about PrEP in health care settings. While participants appreciated the information, counseling and support offered by many health workers, many wished these resources were available in less stigmatizing settings.

*"Sometimes the health care workers are not aware of particular communities. Let me start with the LGBTI community. You find that [the health care workers] do not have enough information. When they see a man wearing a skirt, they chase the person away without knowing what he seeks and sometimes they do not let him access a service, or they skip him and receive other people due to discriminating against him."*

*- 31-year-old male, never on PrEP*

## PrEP protecting both individuals and the community

Among participants currently or previously using PrEP, a major motivator for use was the ease of mind it provided, allowing people to feel more protected from HIV. At the individual level, participants described how they felt less anxious in terms of the number of partners they had, sexual encounters that were unexpected or with unknown partners, and not needing to negotiate condom use. For sex workers in particular, economic benefits - including taking on additional clients and engaging in higher-paying condomless sex - were seen as substantial incentives for taking PrEP. For example:

*"The foremost factor is that it's a protection. The second, it brings inner peace; within this job we tend to worry about the infection but even after unprotected sex you feel safe as long as you're taking the medicine."*

*- 31-year-old female, on PrEP*

*"Most of us sleep with different unfamiliar people and without getting tested. They give us money, and due to life circumstances, you accept. When they ask to have condomless sex, you accept it as well so that your children can eat, but sometimes you take him home and he removes the condom when you did not have an agreement. That is why I take it."*

*- 24-year-old female, previously on PrEP*

A number of participants reflected on the community benefits of PrEP as something that motivated PrEP use. Many described HIV as a serious problem in their communities, one that required an "all hands on deck" |approach to address. In this vein, several GBMSM described how PrEP use would not only benefit themselves, but would benefit sex partners (by indirectly protecting them from HIV), the larger GBMSM community (by reducing community HIV burden), family members (by reducing the potential shame of being associated with an individual living with HIV), and Rwandan society as a whole. GBMSM in particular were highly motivated to encourage others in their network to learn more about and use PrEP so that their entire community could benefit. Some examples include:

*Generally, if someone takes [PrEP], they also protect others from being exposed but when he's infected, they shall infect others. Therefore, taking the protection also protect others.*

*- 31-year-old female, currently on PrEP*

*If my peers took [PrEP] and can prevent themselves [from acquiring HIV], it would be beneficial to me because my peer can sleep with a seropositive individual and if he doesn't use the medication, he would contract it and he can transmit it to me if we sleep together… it forms a cycle but if he uses the medication, it would be a solution to my health.*

*- 31-year-old male, never on PrEP*

*"The country itself benefits from PrEP too, because when the number of people living with HIV reduces, the country regains its capacity. As AIDS is a deadly disease, when the mortality cases reduce, a nation retains manpower; hence, human resources will serve their nation. It reduces disputes and trauma in some families, normally when people inform their family that they're living with HIV, conflicts arise and the family excludes them. Yet, if they are HIV negative…it brings harmony in the family, community and country as a whole."*

*- 29-year-old male, never on PrEP*

**Multi-level stigmas impacting PrEP use for GBMSM and sex workers**

Nearly all participants described HIV-related stigma and fear of discrimination as enormous barriers to PrEP engagement. A dominant theme in interviews was that because there was limited awareness of PrEP in society, and even among some health care providers, PrEP medications were confused with antiretroviral therapy for HIV, and PrEP use would be mistakenly interpreted as having positive HIV status. These fears manifested in various ways: at the interpersonal level, participants described worrying about accidental disclosure of PrEP use if medications were discovered by others; fear of being seen by acquaintances during long waits at a health center, particularly waiting at the HIV program where PrEP is distributed, and being perceived as promiscuous if their PrEP status was inadvertently disclosed. At the structural level, there was anxiety about and even direct experience with poor treatment from healthcare providers. Because of these barriers, both GBMSM and FSW took measures to limit disclosure of PrEP use to

a limited number of people, mostly sex partners, friends and their medical providers. Participants described these barriers:

> "On my part, stigma is also a barrier… when the client comes and says, 'You are seropositive, that's the medication you are taking,' and you respond that it is a medication that prevents contracting HIV but he doesn't believe it and you also don't feel good about it.
>
> - 18 year-old female, stopped PrEP

> "The challenges can be faced by those who use it. The challenges would be to access it rapidly or at a nearby place. Other challenges would be among people who live in families. It can be difficult for them to take it. For them to live with different people, they would discriminate against him, thinking that he is seropositive when he takes a tablet daily."
>
> - 32-year-old male, stopped PrEP

> ''Stigmatization is still existing. If someone who look like a girl, goes at a health center and meet a crowd who is waiting for ART or VCT (Voluntary HIV counselling and testing) they stare and gossip about him. Outpatients, other staff such as cleaners have no training about that only care givers do. So, every time the person goes for the service, he's excluded due to his appearance or identity. Hence saying that stigmatization really exists!"
>
> - 29-year-old male, never on PrEP

For GBMSM, HIV-related stigma intersected with stigma around sexual orientation and gender identity. At the structural level, GBMSM participants reported anticipated and enacted intersectional stigma at health centers, including poor treatment by health care workers. Some described how community health workers living in the same communities as patients, and even medical providers working in health centers, were insufficiently trained on issues of LGBTI health and PrEP, resulting in uncomfortable experiences and explicit discrimination. These mirrored more widespread stigma and discrimination at the societal level towards GBMSM, and made it difficult for some to initiate or continue engaging in PrEP care. As one GBMSM noted:

> ''So I don't think that all doctors all over the country have been trained on LGBTI community; even those who were trained, some of them don't embrace it, consequently they will abuse you. Frankly, if they do not love the community, they'll not serve you or give you bad service in case of consultation. It discourages us and we can't dare to come back again, thinking how bad we are treated as if we're beggars."
>
> - 32-year-old male, stopped PrEP

Participants described other barriers to PrEP use, including competing demands (e.g., work, distance from health center, transportation costs) that made it difficult to attend appointments, side effects leading to discontinuation, or food insecurity that made PrEP a lower priority. For many, not initiating or discontinuing PrEP was not the result of a single factor - rather, the cumulative impact of structural, logistical and stigma-related barriers made the opportunity cost of PrEP use too high. For others, there was sufficient internal or external motivation to overcome stigma, side effects, and other barriers, and continue using PrEP.

## Community access as a critical strategy to improve PrEP engagement

Given the actual and anticipated discomfort and discrimination facing participants, nearly all voiced support for making access to and engagement in PrEP less stigmatizing. Participants in particular expressed their wish to engage in PrEP care in places that were safe and trusted, increase access to information and support, and decrease their exposure to discrimination and related stigma.

Participants advocated for interventions aimed at stigma reduction, such as long-acting PrEP formulations, which would avoid inadvertent disclosure associated with taking daily pills, and changing the packaging of PrEP to clearly differentiate it from antiretrovirals prescribed for HIV. Some participants, particularly GBMSM advocated for PrEP delivery in community settings, which were considered less stigmatizing, safer, more discreet and easier to access, rather than health centers. These included KP organizations, pharmacies, and even one-on-one delivery by community health workers. One participant stated,

*"[Getting PrEP at a community-based, key population association] would reduce the distance covered since it would be close plus in our organization, we understand and know each other, and harassment would not occur, like the challenge I mentioned of finding many people at the health center and as a member of the community you can be afraid to pass there but I think an organization is like home, you cannot be afraid or shy but it would be better if a nurse was available as well to educate us and examine us to know our HIV status before we get the medication."*

*- 28-year-old male, currently on PrEP*

Others preferred receiving PrEP care in health centers, which were better resourced and provided more structured care. FSW were more likely than MSM to highlight logistical barriers to accessing care in health centers and advocate for measures to facilitate access to these settings. While some GBMSM expressed a preference for health center-based care, many felt that additional training of staff on PrEP and in particular, caring for sexual and gender minorities, was essential to ensuring adequate services. Speaking in reference to a health center considered friendly to LGBTI persons, one participant explained,

*"Health centers can be included [as places to access PrEP] but the problem is that not all of them accept LGBT community. So personally, I'd consult those places because I'm very sure that their staff had trainings…"*

*- 24 year-old male, never on PrEP*

## Discussion

In this study of Rwandan GBMSM and FSW, participants described the key role of social networks as sources of PrEP information and highlighted how peers and friends in these networks helped them overcome PrEP-related barriers and provided motivation to engage in care. Our findings suggest that approaches directed at strengthening key population communities and reducing stigma related to HIV and sexual and gender identity could facilitate improved, sustained engagement in PrEP.

We found that for both GBMSM and FSW, community is a central element of PrEP engagement. Key sources of information about PrEP included formal community structures (e.g., associations, peer navigators) as well as less formal, interpersonal friendships and relationships. Participants described a high degree of trust in these sources of information and a corresponding high degree of comfort in accessing them. These findings are similar to research from other SSA settings describing high acceptability of in-person and virtual peer-led communication about PrEP for KPs [37,38], indicating that such approaches can be empowering [39] and even improve engagement in care [40]. Nonetheless, participants in this study described how erroneous or demotivating information can be amplified by community voices in ways that serve as barriers to PrEP. Countering such misinformation through general and targeted dissemination campaigns - as suggested by participants in this study and others [37–38] - could help ensure that these trusted sources provide high-quality information.

Participants reported very high levels of interpersonal and institutional stigma and discrimination related to being misidentified as living with HIV and the potential disclosure of their membership in a key population. Both GBMSM and FSW described HIV-related stigma in their communities (e.g., worry that people would see their medication and think they were using ART) as well as health centers (e.g., being considered promiscuous by health care workers), consistent with findings across SSA [41,42]. Making access to PrEP easier, more discrete, and less fraught - through long-acting injectable formulations [43], low-barrier, community-based access points, and training of providers to provide more welcoming care - was preferred by many participants. For GBMSM, HIV-related stigma was compounded by stigma related to sexual orientation and gender identity, which was felt to be pervasive in society including in many health centers, consistent with national survey data from Rwanda [31]. Because of this, many expressed a desire for PrEP delivery in LGBTI community organizations, similar to preferences identified by GBMSM in other SSA settings [38]. Notably, not all participants felt that community-based PrEP delivery options were ideal, with some preferring more formal and better resourced health settings. Recent data from the SEARCH trial in Uganda and Kenya indicate that providing individuals with choices (e.g., service location) can increase PrEP coverage [44]. While that study did not report on sexual orientation and likely included few GBMSM or FSW, our data suggest that offering options for PrEP care may be a promising strategy.

Most studies of KPs in SSA to date have examined the individual benefits of, facilitators of and barriers to PrEP, yet few have assessed the interpersonal, community aspects of prevention. We observed a high degree of community-focused PrEP motivation among both GBMSM and FSW participants, who identified HIV as a serious problem in their social networks and described how PrEP engagement could benefit sex partners, family members, and peers. These findings echo the concepts of collective antiretroviral protection and prevention solidarity, recently described by Brisson, et al [45]. Much like the idea of herd immunity - where unvaccinated individuals in a population benefit from the vaccination of others - collective protection suggests that when a substantial portion of a community remains HIV-negative (or, if living with HIV, virally suppressed), all members of the community have a reduced HIV risk. Although participants in this study did not explicitly describe the concept of collective antiretroviral protection, our results suggest that some GBMSM and FSW in Rwanda are actively considering how their individual PrEP-related decisions can benefit others around them. Rwandan society has been considered one with a high degree of social cohesion [46], and it is possible that prevention solidarity might have less potential in other settings. Nonetheless, collective HIV prevention has been described among heterosexual, serodifferent couples in Uganda and Kenya, although interpersonal dynamics may differ substantially in these relationships compared to those in broader social networks of GBMSM and FSW [47,48]. To our knowledge, prevention solidarity has not been examined among GBMSM or FSW in SSA. Understanding how to best implement collective antiretroviral protection among GBMSM and FSW in SSA could be a key step to improving and maintaining PrEP engagement.

While analysis revealed many common themes among GBMSM and FSW, we found important differences as well. For GBMSM, anticipated and experienced sexual and gender stigma in society as well as in health centers led to the social network emerging as the single most important source of information, motivation and support related to PrEP engagement. These findings are consistent with prior research we conducted highlighting the close-knit nature of these networks [23], and support recent PEPFAR recommendations to strengthen LGBTI associations to ensure dissemination of accurate PrEP information and facilitate peer linkage to PrEP [30]. FSW, in contrast, described more informal social networks and were more comfortable accessing care in health centers. Many FSW also emphasized economic motivations for PrEP use and highlighted logistical barriers. Together, these findings suggest that efforts to disseminate PrEP information and motivate engagement among GBMSM and FSW should not follow a uniform approach, and to the degree possible should be tailored to the needs of each community, aligned with calls for differentiated approaches to HIV prevention [49].

This study has several limitations. We used a single interview guide for GBMSM and FSW given the many common issues around PrEP access impacting both groups; nonetheless this may have limited our ability to fully explore existing differences between them. We did not recruit persons who had stopped and then restarted PrEP, and thus are not able to

assess how their experiences may differ from persons never, previously or continuing on PrEP. Some interviews occurred in health care settings, and it is possible that participants may have felt reluctant to fully describe their perspectives on health care because of social desirability bias. Willingness to participate in research may reflect an overall lower degree of stigma, and we therefore may not have captured the perspectives of KP who are too stigmatized to enroll in a research study.

In conclusion, we found that for GBMSM and FSW in Rwanda, despite high levels of PrEP awareness in their communities, PrEP engagement remains challenging because of the potential to be misidentified as having HIV and because of stigma related to membership in a key population group. Nonetheless, participants highlighted the strength and support they experienced from peers and colleagues in their social networks, and described a strong motivation to use PrEP as a way to protect both themselves and their communities from HIV. Our findings suggest that leveraging community resources for disseminating information about HIV prevention and delivering PrEP could contribute to successful implementation of PrEP for GBMSM and FSW in Rwanda and other settings in SSA.

## Supporting information

**S1 Text.  In-depth interview guide for study participants.**
(PDF)

**S2 Text.  Questionnaire on inclusivity in global research.**
(DOCX)

## Author contributions

**Conceptualization:** Jonathan Ross, Adebola Adedimeji, Viraj V. Patel.

**Data curation:** Josephine Gasana, Charles Ingabire.

**Formal analysis:** Jonathan Ross, Josephine Gasana, Natalia Zotova, Giovanni Ndabakuranye, Fabiola Mabano, Charles Ingabire.

**Investigation:** Jonathan Ross, Gad Murenzi, Viraj V. Patel.

**Project administration:** Jonathan Ross.

**Supervision:** Adebola Adedimeji, Gad Murenzi.

**Writing – original draft:** Jonathan Ross, Josephine Gasana.

**Writing – review & editing:** Jonathan Ross, Josephine Gasana, Natalia Zotova, Giovanni Ndabakuranye, Fabiola Mabano, Charles Ingabire, Adebola Adedimeji, Gad Murenzi, Viraj V. Patel.

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
