## [Decision Letter · Decision Letter 0]

31 Jan 2025

PGPH-D-24-02716

PrEP in the key population community: a qualitative study of perspectives on pre-exposure prophylaxis by men who have sex with men and female sex workers in Kigali, Rwanda

Dear Dr. Ross,

Thank you for submitting your manuscript to PLOS Global Public Health. After careful consideration, we feel that it has merit but does not fully meet PLOS Global Public Health’s publication criteria as it currently stands. Therefore, we invite you to submit a revised version of the manuscript that addresses the points raised during the review process.

Editor comments:

The reviewers and I found the manuscript to be well-written and offer important insights into the experiences of key populations concerning PrEP in Rwanda. However, there are a few areas where clarifications would strengthen the clarity and impact of the manuscript.I agree with the reviewers' recommendations to ensure that person-centered language is used to refer to key populations throughout. Please also note and address the questions concerning qualitative data collection and participant recruitment, and consider whether these aspects might need to be discussed as study limitations.In addition to the reviewers' comments, I would encourage the authors to expand the background and introduction to provide more information on the context of key populations in Rwanda and the policy/legal environment. In the discussion, the paper references that the study city has a "relatively tolerant policy climate with respect to sexual health", but it isn't clear what that means or how that might play a role in interpreting findings, beyond being cited as a limitation. Additionally, given that the goal of qualitative studies is not generalizability, but instead an in-depth understanding of experiences and context, it isn't clear to me how this aspect would necessarily constitute a limitation.

We look forward to receiving your revised manuscript.

Kind regards,

Marie A. Brault, PhD

Academic Editor

Journal Requirements:

**Please only choose the relevant sentences from below**

i. Please clarify all sources of funding (financial or material support) for your study. List the grants (with grant number) or organizations (with url) that supported your study, including funding received from your institution. 

ii. State the initials, alongside each funding source, of each author to receive each grant.

iii. State what role the funders took in the study. If the funders had no role in your study, please state: “The funders had no role in study design, data collection and analysis, decision to publish, or preparation of the manuscript.”

iv. If any authors received a salary from any of your funders, please state which authors and which funders.

3. We have amended your Competing Interest statement to comply with journal style. We kindly ask that you double check the statement and let us know if anything is incorrect. 

4. In the online submission form, you indicated that "The datasets analyzed during the current study are available from the corresponding author on reasonable request.". 

3. Uploaded as supplementary information.

5. We have noticed that you have uploaded Supporting Information files, but you have not included a list of legends. Please add a full list of legends for your Supporting Information files after the references list. 

Additional Editor Comments (if provided):

Reviewers' comments:

Reviewer's Responses to Questions

**Comments to the Author**

1. Does this manuscript meet PLOS Global Public Health’s publication criteria ? Is the manuscript technically sound, and do the data support the conclusions? The manuscript must describe methodologically and ethically rigorous research with conclusions that are appropriately drawn based on the data presented.

Reviewer #1: Yes

Reviewer #2: Yes

Reviewer #3: Yes

Reviewer #4: Yes

2. Has the statistical analysis been performed appropriately and rigorously?

Reviewer #1: Yes

Reviewer #2: N/A

Reviewer #3: Yes

Reviewer #4: Yes

3. Have the authors made all data underlying the findings in their manuscript fully available (please refer to the Data Availability Statement at the start of the manuscript PDF file)?

Reviewer #1: Yes

Reviewer #2: Yes

Reviewer #3: Yes

Reviewer #4: Yes

4. Is the manuscript presented in an intelligible fashion and written in standard English?

Reviewer #1: Yes

Reviewer #2: Yes

Reviewer #3: Yes

Reviewer #4: Yes

5. Review Comments to the Author

Reviewer #1: This well-executed and written study describes the qualitative barriers and facilitators of PrEP use among men who have sex with men and female sex workers in Kigali. It provides useful comparisons of different motivators and barriers, and includes an interesting discussion of social cohesion and importance of collective benefits of PrEP in this setting. It is a valuable addition to the literature and only minor edits are suggested.

Pg. 3, line 87: instead of “high-risk groups”, prioritise using people-centred language that doesn’t define people solely by their risk. e.g. “groups with higher risk of acquiring HIV”.

Pg. 3, line 88: Is the number receiving PrEP medication based on having received it at least once? Are there any data on PrEP continuation that could also be cited here? Not knowing the population size estimates of MSM and FSW in Rwanda/Kigali, is there any estimate of the proportion of these groups who are on or who have started PrEP?

Pg. 8, line 252: Is colleague the correct translation here, referring to coworker ?

Pg. 9, line 291: Based on some of the included quotes, also implies stigma experienced around gender identity and expression.

Reviewer #2: This well-written manuscript presents a qualitative analysis of cisgender female sex workers and gay, bisexual, and other men who have sex with men in Kigali, Rwanda. While several findings have been previously published, and it would have been preferable to avoid the comingling of qualitative data from these two distinct populations—given that existing research highlights differences in their health experiences—the paper is noteworthy for exploring “collective antiretroviral protection” as a sexual health strategy. A strength of the study is that participants were recruited from facility and community settings and included current, former, and never PrEP users. However, the PrEP perspectives of never-users are well described in the literature; thus, including PrEP re-initiators would have been more interesting, given how users cycle on and off PrEP. Given that intersecting stigmas are recognized as a critical and complex barrier to the uptake of evidence-based HIV prevention interventions, it would have been beneficial to organize the Results (third theme) and Discussion in line with the Socio-Ecological Model, which the authors indicate they referenced in designing the interview guide. I have a few comments which the authors might find helpful.

1) Abstract and Introduction—The authors state that “limited data exist describing experiences and preferences of GBMSM and FSW with respect to accessing and using PrEP” as a rationale for their study. However, a PubMed search (MSM AND PrEP AND Qualitative AND Africa) revealed 23 studies, suggesting their PrEP experiences have been amply described in the literature.

2) Introduction and Methods: Please clarify that this is a study of oral PrEP, given the availability of PrEP rings and injections as alternative formulations.

3) Methods: Participants were recruited from June-July 2022 but interviewed from November-December 2022. What accounts for this five-month gap? Did all who indicated interest in the study eventually participate in interviews, or did some drop off or could not be reached five months later? Were all participants who were approached interviewed, or did some decline? Reasons for refusal should be described in the Results.

4) Methods: Line 111 refers to an interview guide. Were separate interview guides used for FSW and GBMSM? If not, this should be acknowledged as a limitation as these are distinct sub-populations.

5) Results, Table 1. Six GBMSM reported sex with both women and men, but marital status was not reported. Were they married to women to fulfill cultural and religious norms to have children and to avoid discrimination, as often happens in this setting?

6) Results, lines 163-170: This text is unrelated to the social networks theme and can be moved to after the sentence ending on line 159.

7) Results–Quantification of qualitative research. It is tempting to describe qualitative data using quantitative terms, e.g., “a high level of knowledge about PrEP” (how was this measured?), “a substantial number of participants,” or “tremendous amount of anticipated and enacted stigma” (how was this measured?) but this is best avoided.

8) Discussion: The authors state in the Methods that the socioecological model was used to explore individual and structural PrEP barriers and facilitators. However, this is not evident in the Results (lines 352-367), which do not appear to discuss the findings through the lens of individual, interpersonal, intersectional, or structural stigma.

9) Discussion: Whereas stigma and discrimination in healthcare settings were described in the Results, there is no mention of gender sensitivity training for providers as a possible solution. Additionally, the paper does not mention sex work stigma (sex work is criminalized in Rwanda), particularly for GBMSM who sold sex, as per Table 1.

10) Discussion: Generalizability – avoid the temptation to generalize findings from FSW and GBMSM in one urban setting by saying your “data suggest that offering options for PrEP care would be well received by key populations in SSA.”

11) Language: The NIAID HIV Language Guide recommends not using the term MSM because it is stigmatizing. Gay, bisexual, and other men who have sex with men (GBMSM) is preferred. Moreover, Table 1 indicates that some of these men were bisexual, which speaks to the limitations of describing this population as MSM.

Other comments

12) Line 59: Clarify that the population under study is cisgender FSW and GBMSM. Currently, the reader only finds this out after seeing sex at birth data in Table 1.

13) Lines 87-88: FSW and MSM were defined in the Introduction and do not need to be written in full in the Methods.

14) Lines 148-149: Please provide a citation for the COREQ guidelines.

15) Line 382: The preferred term is “serodifferent” not serodiscordant.

Reviewer #3: A well-written manuscript with a good review of literature.

Few comments:

- In qualitative research, the interviewer is the instrument. You have mentioned that three researchers were trained (?) on qualitiative research. Those interviewers need to be well versed with exploration and proping of issues and potential themes.

- I have concerns about the emerging themes. Who decided to stop at this number of participants? Generally, we are looking for saturation in emerged themes before we decide to stop further recruitment. I cannot see any discussion about this.

- Would be good if you present information about recruitment- i.e. how many approached and how many refused to participate. This will give an additional idea about enrolled individuals- as you highlighted mught be very motivated. You also need to present information about recruitment of individuals from each recruitment centre, the three medical centre and 7 community centres.

Reviewer #4: Thank you for the opportunity to review this draft manuscript.

Overall, the paper is clear and well-written and was enjoyable to read.

Background

-Section that states that current care models are not optimized for KPs can be expanded to be more descriptive of what current care models there are and why they are not working well, or describe that they are tailored towards other risk groups. The evidence described on factors does not explicitly speak to this. But this implies that PrEP services are not differentiated to the population, for example primarily being clinic/health facility based. But some description of this and references would be helpful.

-Suggest including a citation for the government of Rwanda initiation of PrEP for KPs (first line of 3rd paragraph)

-For context for the reader, in the background or methods, could indicate if same-sex sexual activity is considered legal in Rwanda, as well as if sex work is legal or criminalized in the country. This may be influential in terms of how individuals access care/stigma felt, etc. and this information would be helpful for the reader.

Methods

-Is there any population estimate for FSW and MSM – in Rwanda overall, or in Kigali? This will be helpful, as list the number of individuals on PrEP as of June 2023.

-FSW and MSM are spelled out in paragraph 2 of the Methods – as acronym has already been used, can use consistently throughout paper

-In the methods, it is indicated that individuals were purposefully recruited to include those that were using PrEP, had previously used PrEP, and those that had never used PrEP. Is there a eligibility screening tool that was used that can be provided as part of the supplementary materials? The in-depth interview guide provided asks questions including ‘Have you ever heard of PrEP?’, but to be recruited, it seems like this question would already have been asked/known? Same with have you ever used PrEP question.

-Please review the interview guide provided. In one place it indicates in Section II, Question 2, have you ever used PrEP, if no, ask 2a, and skip to Section II. I think this means skip to Section III. Please review and revise as needed.

Results

T-he time between recruitment and interviews was long (several months). Were any individuals recruited lost to follow-up because of this delay between recruitment and interview?

-Spell out LGBTI at first use

-There are a number of yes/no questions in the interview guide – have you considered adding these to the results?

-Additional quotations from participants in the stigma section of the results could be beneficial. In particular, in this section there is not a quote from a FSW.

-Additional quotes in the community access results section would be beneficial to the paper.

-Overall, the FSW point of view in the results seems limited. Were there any differences by KP group in terms of how delivery of PrEP is preferred (i.e., in the community or at a health center, delivery at home by a community health care workers…?). I see now that this is mentioned in a paragraph in the discussion – emphasizing the importance of differentiated care, however, it would be beneficial to include these findings in the results, as that is not clear currently.

Discussion

-Some of the probes used in the interview guide seem leading – i.e., biggest challenges accessing PrEP. Did the probes influence the participants’ responses in any way? Is this a potential limitation to add?

-As the study made sure to include individuals who were currently on PrEP, had stopped PrEP, had never started PrEP, were the data analyzed by these disaggregations in any way?

6. PLOS authors have the option to publish the peer review history of their article (what does this mean? ). If published, this will include your full peer review and any attached files.

**Do you want your identity to be public for this peer review?** For information about this choice, including consent withdrawal, please see our Privacy Policy .

Reviewer #1: No

Reviewer #2: No

Reviewer #3: **Yes: ** Haider Al-Darraji

Reviewer #4: No

---

## [Decision Letter · Decision Letter 1]

3 Apr 2025

PrEP in the key population community: a qualitative study of perspectives on pre-exposure prophylaxis by gay, bisexual and other men who have sex with men and female sex workers in Kigali, Rwanda

PGPH-D-24-02716R1

Dear Dr. Ross,

We are pleased to inform you that your manuscript 'PrEP in the key population community: a qualitative study of perspectives on pre-exposure prophylaxis by gay, bisexual and other men who have sex with men and female sex workers in Kigali, Rwanda' has been provisionally accepted for publication in PLOS Global Public Health.

Best regards,

Marie A. Brault, PhD

Academic Editor

Reviewer Comments (if any, and for reference):

Reviewer's Responses to Questions

**Comments to the Author**

1. If the authors have adequately addressed your comments raised in a previous round of review and you feel that this manuscript is now acceptable for publication, you may indicate that here to bypass the “Comments to the Author” section, enter your conflict of interest statement in the “Confidential to Editor” section, and submit your "Accept" recommendation.

Reviewer #2: All comments have been addressed

Reviewer #4: All comments have been addressed

2. Does this manuscript meet PLOS Global Public Health’s publication criteria ? Is the manuscript technically sound, and do the data support the conclusions? The manuscript must describe methodologically and ethically rigorous research with conclusions that are appropriately drawn based on the data presented.

Reviewer #2: Yes

Reviewer #4: Yes

3. Has the statistical analysis been performed appropriately and rigorously?

Reviewer #2: N/A

Reviewer #4: N/A

4. Have the authors made all data underlying the findings in their manuscript fully available (please refer to the Data Availability Statement at the start of the manuscript PDF file)?

Reviewer #2: No

Reviewer #4: No

5. Is the manuscript presented in an intelligible fashion and written in standard English?

Reviewer #2: Yes

Reviewer #4: Yes

6. Review Comments to the Author

Reviewer #2: (No Response)

Reviewer #4: Thank you for responding to all reviewer comments and incorporating feedback into the manuscript.

7. PLOS authors have the option to publish the peer review history of their article (what does this mean? ). If published, this will include your full peer review and any attached files.

**Do you want your identity to be public for this peer review?** For information about this choice, including consent withdrawal, please see our Privacy Policy .

Reviewer #2: No

Reviewer #4: No
